# Neurohormonal signaling via a sulfotransferase antagonizes insulin-like signaling to regulate a *Caenorhabditis elegans* stress response

Nicholas O. Burton[1,2], Vivek K. Dwivedi[1], Kirk B. Burkhart[1], Rebecca E.W. Kaplan[3], L. Ryan Baugh [3] & H. Robert Horvitz[1]

Insulin and insulin-like signaling regulates a broad spectrum of growth and metabolic responses to a variety of internal and environmental stimuli. For example, the inhibition of insulin-like signaling in *C. elegans* mediates its response to both osmotic stress and starvation. We report that in response to osmotic stress the cytosolic sulfotransferase SSU-1 antagonizes insulin-like signaling and promotes developmental arrest. Both SSU-1 and the DAF-16 FOXO transcription factor, which is activated when insulin signaling is low, are needed to drive specific responses to reduced insulin-like signaling. We demonstrate that SSU-1 functions in a single pair of sensory neurons to control intercellular signaling via the nuclear hormone receptor NHR-1 and promote both the specific transcriptional response to osmotic stress and altered lysophosphatidylcholine metabolism. Our results show the requirement of a sulfotransferase–nuclear hormone receptor neurohormonal signaling pathway for some but not all consequences of reduced insulin-like signaling.

[1] Howard Hughes Medical Institute, Department of Biology, Massachusetts Institute of Technology, Cambridge, MA 02139, USA. [2] Centre for Trophoblast Research, Department of Physiology, Development and Neuroscience, University of Cambridge, Cambridge CB2 3EG, UK. [3] Department of Biology, Duke University, Durham, NC 27708, USA. Correspondence and requests for materials should be addressed to N.O.B. (email: nob20@cam.ac.uk) or to H.R.H. (email: horvitz@mit.edu)

Changes in insulin and insulin-like growth factor signaling can elicit distinct responses in different cellular or developmental contexts. For example, obese humans exhibit a transcriptional response to insulin in adipose tissue that differs from that of non-obese individuals[1,2]. This response is independent of plasma insulin concentration[1], indicating that the consequences of insulin signaling are distinct in different physiological contexts. Similarly, the consequences of insulin signaling can change during development. For example, injection of the insulin-like peptide Bombyxin-II into hemolymph of larvae but not into hemolymph of adults of the silk moth *Bombyx mori* results in increased conversion of trehalose into glucose[3,4].

Although the specific physiological responses to changes in insulin signaling vary among animals, the general ability to respond to insulin signaling and to modify the specificity of responses to insulin signaling as a consequence of changing environmental and developmental stimuli is conserved among metazoa. An understanding of the molecular mechanisms that govern the sensitivity and specificity of insulin signaling in different contexts is essential for the understanding of animal development and physiology and likely to prove important for the understanding of pathologies that result from abnormalities in insulin signaling, such as obesity and type 2 diabetes.

We previously reported that *C. elegans* arrests development immediately after hatching in response to osmotic stress and that this developmental arrest involves the inhibition of insulin-like signaling and subsequent activation of the FOXO transcription factor DAF-16[5]. Earlier studies had showed that *C. elegans* also arrests development immediately after hatching in response to starvation and that this developmental arrest also involves the inhibition of insulin-like signaling and activation of DAF-16[6]. Despite these similarities in the timing and regulation of developmental arrest in response to osmotic stress and starvation, we found that these two developmental arrests are physiologically distinct[5]. Specifically, (a) developmental arrest in response to osmotic stress results in animals that are immobile and do not respond to touch, whereas animals arrested in response to starvation remain mobile and responsive, and (b) most genes that exhibit changes in expression in animals arrested in response to osmotic stress do not exhibit changes in expression in animals arrested in response to starvation and vice versa[5]. These results indicate that the inhibition of insulin-like signaling in *C. elegans* can elicit distinct physiological responses to starvation and osmotic stress. The mechanistic basis of these different responses has been unknown. Here we demonstrate that the cytoplasmic sulfotransferase SSU-1 is required in the ASJ sensory neurons for developmental arrest in response to osmotic stress, but not in response to starvation, and functions by controlling neurohormonal signaling via the nuclear hormone receptor NHR-1 to antagonize insulin-like signaling.

## Results

### Identification of SSU-1 as a regulator of developmental arrest.

To determine how the inhibition of insulin-like signaling can result in distinct states of arrested development in response to osmotic stress and starvation, we screened approximately 20,000 F3 progeny of EMS-mutagenized animals for mutants that failed to arrest development in response to osmotic stress (500 mM NaCl). A nonsense allele of the cytosolic sulfotransferase gene *ssu-1* (*n5888*, W284Stop) resulted in approximately 40% of animals failing to arrest development in response to 500 mM NaCl (Fig. 1a, b). Similarly, six other independently isolated mutations in *ssu-1* (*fc73, tm1117, gk266317, gk747222, gk876992, gk319712*) caused animals to fail to arrest development in response to osmotic stress (Fig. 1b). To test if SSU-1 is also required for developmental arrest in response to starvation, we starved wild-type, *daf-16* and *ssu-1* mutant animals for 1 week and assayed the

percentage of animals with a divided M cell; M-cell division does not occur in wild-type animals that arrest development in response to starvation[6]. We found that 0% of M cells in both wild-type animals and *ssu-1* mutants failed to arrest cell division (Fig. 1c). By contrast, 8% of M cells in *daf-16* mutants failed to arrest cell division in response to starvation (Fig. 1c), consistent with previous observations[6]. Thus, SSU-1 is required for developmental arrest in response to osmotic stress but is not required for developmental arrest in response to starvation.

SSU-1 is expressed in a single pair of sensory neurons, the ASJ neurons[7]. To determine if SSU-1 functions in the ASJ sensory neurons to regulate developmental arrest in response to osmotic stress, we expressed a rescuing transgene of *ssu-1(+)* under the control of the ASJ-specific promoter *trx-1* in *ssu-1(-)* animals[8]. Expression of SSU-1 in the ASJ sensory neurons restored developmental arrest in response to osmotic stress (Fig. 1d), indicating that SSU-1 functions in the ASJ sensory neurons.

### SSU-1 controls signaling via NHR-1.

In humans, the cytosolic sulfotransferases sulfonate steroid hormones such as dehydroepiandrosterone (DHEA) and pregnenolone[9]. These hormones regulate gene expression by activating nuclear hormone receptors[9]. We hypothesized that SSU-1 might similarly regulate the sulfonation of a hormone that promotes developmental arrest in response to osmotic stress by controlling the transcriptional response to osmotic stress. To determine if SSU-1 regulates gene expression in response to osmotic stress, we exposed wild-type and *ssu-1* mutant embryos to either 50 mM or 500 mM NaCl for 3 h and quantified mRNA expression using RNA-seq. We found that the expression of 434 genes was upregulated greater than twofold in response to osmotic stress and that SSU-1 function was required for the expression of 106 of these genes (Supplementary Dataset 1), including 20 of the 25 genes that exhibited a greater than 10-fold increase in expression in response to osmotic stress ($p < 0.05$, two-tailed *t*-test) (Fig. 2a). For example, the genes that encode the superoxide dismutase *sod-5*[10] and the osmotic stress resistance protein *lea-1*[11] exhibited a >25-fold increase in expression in response to osmotic stress, and their increased expression in response to osmotic stress required SSU-1 (Fig. 2a). We confirmed that SSU-1 was required for the increased expression of *sod-5* in response to osmotic stress using a GFP reporter (Fig. 2b, c). GFP was expressed broadly throughout the animal in response to osmotic stress, and this broad increase in expression required SSU-1 function in the ASJ sensory neurons (Fig. 2b, c). We note that previous studies demonstrated that several genes that exhibit increased expression in response to osmotic stress, such as *gpdh-1* and *gpdh-2*, are required for survival and development in response to osmotic stress[5,12]. It thus seems likely that other genes upregulated in response to osmotic strength, such as *sod-5* and *lea-1*, similarly act to aid survival in response to osmotic stress.

To identify factors that function downstream of SSU-1 to control the transcriptional response to osmotic stress, we performed a larger screen of approximately 200,000 F3 progeny of EMS-mutagenized animals for mutants that both (1) failed to arrest development in response to osmotic stress, and (2) failed to express *sod-5::gfp* in response to osmotic stress. We identified five alleles of the nuclear hormone receptor gene *nhr-1* (*n6217, n6219, n6228, n6231, n6242*) (Fig. 2d, e).

We attempted to determine whether *nhr-1* acts cell-autonomously by testing for rescue of the loss of *sod-5::GFP* expression in *nhr-1* mutants. We expressed a copy of the wild-type *nhr-1* gene using either the endogenous *nhr-1* promoter or any of a variety of cell-type-specific promoters, but we were unable to observe rescue, possibly because of toxic consequences

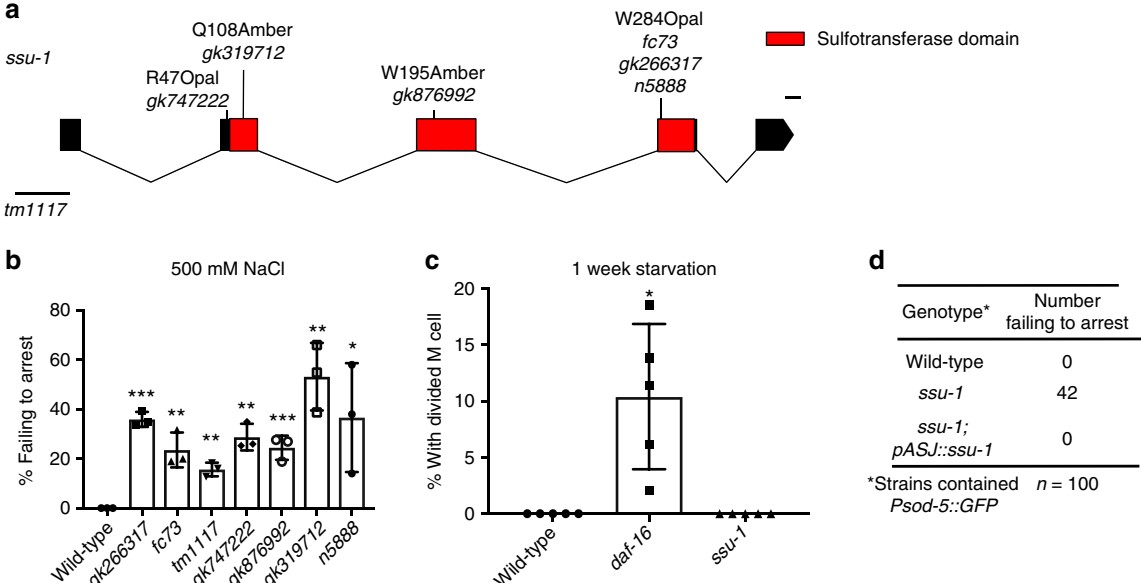

**Fig. 1** The cytosolic sulfotransferase SSU-1 functions in the ASJ sensory neurons to regulate developmental arrest in response to osmotic stress. **a** Schematic of *ssu-1* mutations that cause defects in developmental arrest in response to osmotic stress. Scale bar, 100 bp. Numbers represent amino acid numbers. **b** Percent of *ssu-1* mutants failing to arrest development in response to 500 mM NaCl. *n* = 3 experiments, each with more than 100 animals. Error bars, s.d. **c** Percent of wild-type, *daf-16(mu86)*, and *ssu-1(fc73)* animals with a divided M cell after 1 week of starvation. Animals contained ayIs7 [hlh-8::GFP fusion + dpy-20(+)] to facilitate M-cell identification. *n* > 200 animals. Error bars, s.e.m. **d** Number of wild-type, *ssu-1(fc73)*, and *ssu-1(fc73); ssu-1(+)* animals failing to arrest development in response to osmotic stress. The *trx-1* promoter was used to drive ASJ cell-specific expression of *ssu-1*. Animals also contained wuIs57 [pPD95.77 Psod-5::GFP, rol-6(su1006)], which drives GFP expression in response to osmotic stress. *n* = 100 animals. *\*p* < 0.05, *\*\*p* < 0.01, *\*\*\*p* < 0.001. Two-tailed *t*-test. Source data are provided as a Source Data file

of *nhr-1* overexpression (see Supplementary Discussion). Besides arresting development in response to food deprivation or high osmolarity, *C. elegans* also arrests its development in response to harsh environmental conditions, entering an alternative third-larval stage known as the dauer larva[13]. Like the other developmental arrests, dauer arrest is regulated by insulin-like signaling[13]. Interestingly, we found that *sod-5::GFP* was expressed in dauer-arrested animals and that this expression required NHR-1, as in animals that arrest in response to osmotic stress (Fig. 2f). We found that *sod-5::GFP* expression was cell-autonomously restored in transgenic animals that tissue-specifically expressed a wild-type copy of *nhr-1*, i.e., dauers that expressed *nhr-1* in the intestine expressed *sod-5::GFP* only in the intestine, dauers that expressed *nhr-1* in neurons expressed *sod-5::GFP* only in neurons, and dauers that expressed *nhr-1* in muscles expressed *sod-5::GFP* only in muscles (Fig. 2f). We conclude that NHR-1 functions cell-autonomously to promote gene expression.

Overexpression of *ssu-1* in the ASJ sensory neurons was sufficient to drive the expression of *sod-5::gfp* even in the absence of osmotic stress (Fig. 2g). To test if the expression of *sod-5::gfp* in animals that overexpressed *ssu-1* required NHR-1, we examined *nhr-1* mutants that overexpressed *ssu-1*. Overexpression of *ssu-1* did not result in *sod-5::gfp* expression in *nhr-1* mutants (Fig. 2g). These results indicate that NHR-1 is required for SSU-1 to drive the expression of *sod-5* and suggest that SSU-1 and NHR-1 function in the same pathway. We propose that in the ASJ sensory neurons SSU-1 sulfonates a small-molecule substrate to generate a ligand for NHR-1 and that this ligand then activates NHR-1 in one or more target tissues to drive the transcriptional response to osmotic stress and developmental arrest. In addition, our results indicate that while signaling via SSU-1 is not required for all developmental arrests regulated by insulin-like signaling (Fig. 1c), SSU-1 might regulate aspects of other arrested states, such as the expression of *sod-5* in dauer-arrested animals (Fig. 2f).

**SSU-1 acts antagonistically to insulin-like signaling**. The developmental arrest of *C. elegans* in response to osmotic stress involves the inhibition of insulin-like signaling, which results in the activation of the FOXO transcription factor DAF-16[5]. To determine if DAF-16 and SSU-1 regulate the expression of the same or different target genes, we exposed wild-type and *daf-16* mutant embryos to 50 mM or 500 mM NaCl and quantified mRNA expression by RNA-seq. We found that 161 genes both exhibited a greater than twofold increase in expression in response to osmotic stress and were dependent on DAF-16 for their induction (Fig. 3a, b and Supplementary Dataset 2). Notably, all 25 of the genes that exhibited the largest DAF-16-dependent increases in expression in response to osmotic stress also required SSU-1 for their expression. Similarly, all 25 of the genes that exhibited the largest SSU-1-dependent increase in expression in response to osmotic stress also required DAF-16 for their expression (Supplementary Dataset 2). In total, of the 161 genes regulated by DAF-16 in response to osmotic stress, 64 were also regulated by SSU-1 (Fig. 3c). However, a few genes, such as *gpdh-1*, which exhibited an approximately 50-fold increase in response to osmotic stress, did not require either SSU-1 or DAF-16 for their expression (Figs. 2a, 3a). These results indicate that most but not all of the genes that exhibited the greatest increases in expression in response to osmotic stress required both SSU-1 and DAF-16 for their expression.

Loss-of-function mutations in the insulin-like receptor gene *daf-2* cause animals to be more likely to arrest development in response to osmotic stress than are wild-type animals[5]. To examine interactions among *ssu-1*, *nhr-1*, and *daf-2*, we constructed both *daf-2; ssu-1* and *daf-2; nhr-1* double-mutant animals and exposed these mutants to either mild (300 mM NaCl) or strong (500 mM NaCl) osmotic stress. Consistent with our previous findings, nearly 100% of *daf-2* mutant embryos arrested development at 300 mM NaCl (Fig. 3d). By contrast,

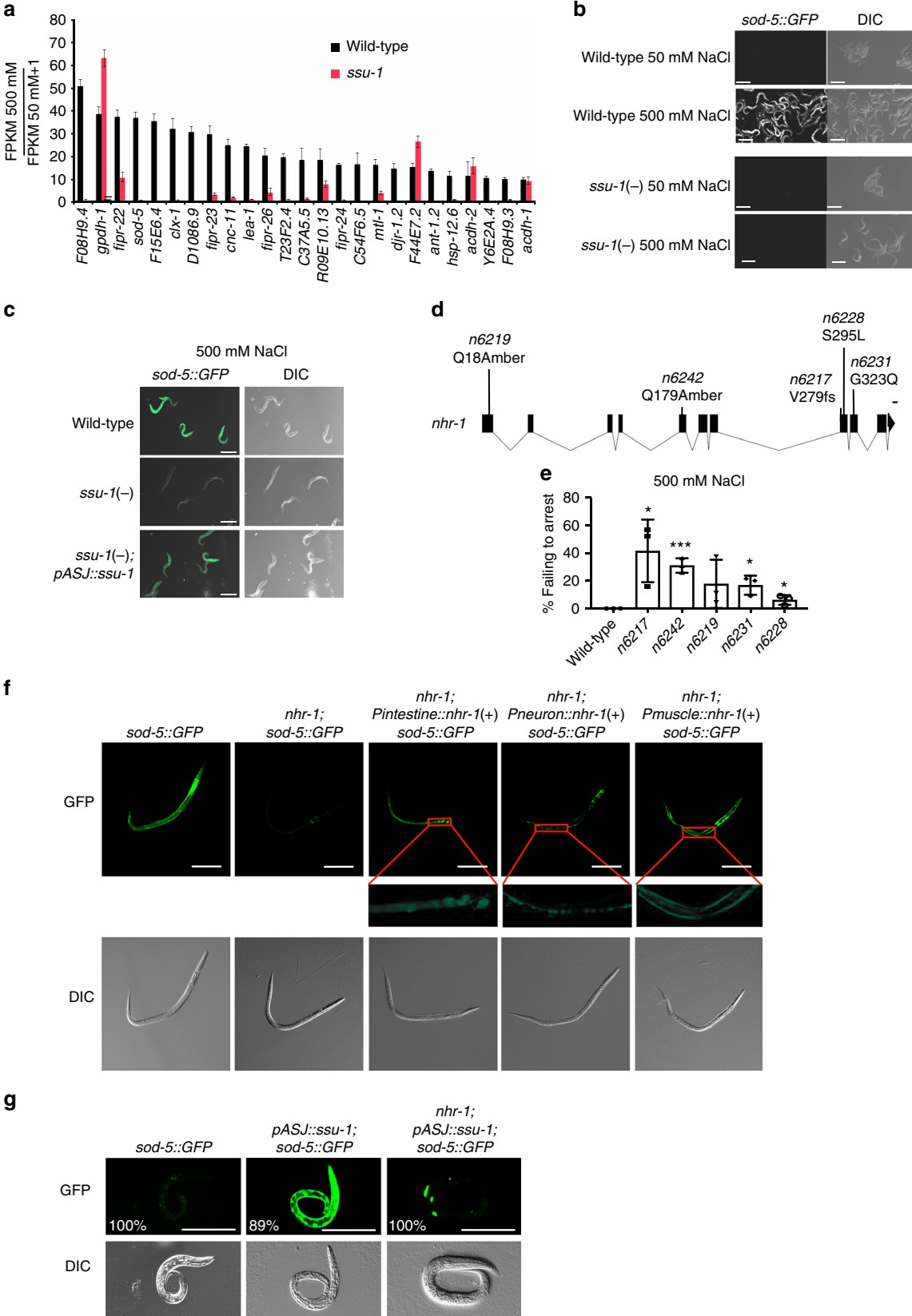

**Fig. 2** The nuclear hormone receptor NHR-1 is required for SSU-1 to promote the transcriptional response to osmotic stress. **a** Average fold change in mRNA expression in wild-type and *ssu-1(fc73)* mutant embryos at 500 mM NaCl as compared to 50 mM NaCl measured by RNA-seq (FPKM 500 mM NaCl/FPKM 50 mM NaCl + 1). Shown are the 25 genes that exhibited the greatest increase in expression in wild-type animals at 500 mM NaCl vs. 50 mM NaCl. $n = $ 3 replicates. Error bars, s.d. **b** Confocal and differential interference contrast (DIC) images showing *Psod-5::gfp* expression in wild-type and *ssu-1(fc73)* L1-stage mutants exposed to 500 mM NaCl for 24 h. Scale bars, 100 μm. **c** Confocal and DIC images of *Psod-5::GFP* expression in wild-type, *ssu-1(fc73)*, and *ssu-1(fc73); ssu-1( + )* L1-stage mutants exposed to 500 mM NaCl for 24 h. The *trx-1* promoter was used to drive ASJ cell-specific expression of SSU-1. Scale bars, 100 μm. **d** Schematic of *nhr-1* mutations that cause defects in developmental arrest in response to osmotic stress. Scale bar, 100 bp. fs: frameshift. Numbers represent amino acid numbers. **e** Percent of *nhr-1* mutants failing to arrest development in response to 500 mM NaCl. $n = $ 3 experiments, each with more than 100 animals. Error bars, s.d. **f** Representative confocal and DIC images of *Psod-5::GFP* expression in wild-type and *nhr-1(n6242)* mutant dauers. The *ges-1* promoter was used to drive intestine-specific expression, the *rab-3* promoter was used to drive neuron-specific expression, and the *unc-54* promoter was used to drive muscle-specific expression. Scale bar, 100 μm. **g** Representative confocal and DIC images of *Psod-5::GFP* expression in wild-type and *nhr-1 (n6242)* L1-stage animals. The *trx-1* promoter was used to drive ASJ cell-specific expression of SSU-1. Animals expressed GFP specifically in coelomocytes under the control of the *unc-122* promoter as a co-injection marker. %, percent of animals that expressed GFP as in the representative image shown. Scale bar, 100 μm. $*p < 0.05$, $**p < 0.01$, $***p < 0.001$. Two-tailed *t*-test. Source data are provided as a Source Data file

none of the *daf-2; ssu-1* mutants and only 30% of the *daf-2; nhr-1* mutants arrested development at 300 mM NaCl (Fig. 3d). These data indicate that both SSU-1 and NHR-1 are required for reduced insulin-like signaling to promote developmental arrest in response to osmotic stress.

At 500 mM NaCl approximately 40% of both *ssu-1* and *daf-2; ssu-1* animals failed to arrest development (Fig. 3e), indicating that SSU-1 is also required for reduced insulin-like signaling to promote developmental arrest in response to 500 mM NaCl. By contrast, approximately 20% of *nhr-1* mutants failed to arrest development at 500 mM NaCl, but all of the *daf-2; nhr-1* double mutants arrested development at 500 mM NaCl (Fig. 3e). These results indicate that NHR-1 is likely not the only downstream effector of SSU-1. In humans, individual sulfotransferases commonly sulfonate multiple targets[9]. Our data suggest that similar to cytosolic sulfotransferases in other organisms, SSU-1 modifies the activities of multiple downstream effectors, including NHR-1, to control the response to osmotic stress.

Insulin-like signaling inhibits the activation of the FOXO transcription factor DAF-16 by sequestering DAF-16 in the cytoplasm[14]. The loss of insulin-like signaling in response to osmotic stress causes developmental arrest because DAF-16 is no longer sequestered in the cytoplasm and can translocate into the nucleus[5]. To determine if SSU-1 signaling similarly promotes the translocation of DAF-16 into the nucleus in response to osmotic stress, we examined wild-type and *ssu-1* mutant animals that expressed a GFP-tagged copy of DAF-16 and assayed DAF-16 translocation to the nucleus in response to osmotic stress. SSU-1 was not required for DAF-16 translocation into the nucleus in response to osmotic stress (Fig. 3f and Supplementary Fig. 1). This observation suggests that signaling via SSU-1 functions in parallel to insulin-like signaling and DAF-16 translocation into the nucleus to regulate development and gene expression in response to osmotic stress.

**SSU-1 and insulin-like signaling regulate metabolism.** We previously found that increases in the levels of glycerol, an osmolyte that protects animals from the effects of osmotic stress[12], can prevent *C. elegans* from undergoing developmental arrest in response to osmotic stress[5]. Since *ssu-1* and *daf-16* mutants fail to undergo developmental arrest in response to osmotic stress, we asked if these mutants have increased levels of glycerol. We used mass spectrometry to profile 137 polar metabolites and 1069 lipid metabolites in wild-type, *ssu-1*, and *daf-16* mutant embryos grown in normal osmotic conditions (50 mM NaCl) (Supplementary Dataset 3). We did not observe any increase in the level of glycerol (Fig. 4a), suggesting that it is not increased glycerol production that prevents *ssu-1* and *daf-16* mutants from undergoing developmental arrest in response to

osmotic stress. A few polar and lipid molecules were slightly changed in both *ssu-1* and *daf-16* mutant embryos ($p < 0.05$, two-tailed *t*-test) (Fig. 4a, b). None of the metabolites we profiled exhibited a greater than twofold change in abundance, suggesting that under normal osmotic conditions the loss of either SSU-1 or DAF-16 function has minimal effects on metabolism.

SSU-1 and DAF-16 might regulate the production of glycerol in response to osmotic stress but not under conditions of normal osmolarity. To determine if SSU-1 and DAF-16 regulate metabolism in response to osmotic stress, we profiled polar and lipid metabolites in embryos exposed to 300 mM NaCl. We found that *ssu-1* and *daf-16* mutant embryos did not produce more glycerol than wild-type embryos in response to osmotic stress (Supplementary Dataset 3), suggesting that SSU-1 and DAF-16 do not regulate glycerol metabolism.

Among the other 1205 metabolites profiled, six polar metabolites were increased in abundance more than twofold ($p < 0.01$, two-tailed *t*-test) in wild-type animals in response to osmotic stress, including glycerol, and 12 polar metabolites that decreased in abundance more than twofold ($p < 0.01$, two-tailed *t*-test) in wild-type animals in response to osmotic stress, including several TCA-cycle intermediates (Supplementary Dataset 3). Of these 18 polar metabolites that changed in abundance in response to osmotic stress, none was regulated by both SSU-1 and DAF-16 (Supplementary Dataset 3). In addition, we identified 63 lipid metabolites that exhibited a greater than twofold decrease in abundance in response to osmotic stress and 85 lipid metabolites that exhibited a greater than twofold increase in abundance in response to osmotic stress. Of these 148 lipid metabolites that changed in abundance, six were regulated by both SSU-1 and DAF-16; strikingly, all six were lysophosphatidylcholines (LPCs) (Fig. 4c). Specifically, six LPCs exhibited between two- and eightfold increases in abundance in response to osmotic stress, and this increase in abundance required both SSU-1 and DAF-16 (Fig. 4c). Thus, embryonic metabolism is affected by osmotic stress, and SSU-1 and DAF-16 are required for the increase in LPC abundance in response to osmotic stress.

We tested whether increased LPC abundance can drive developmental arrest by adding 1 mM of either mixed LPCs from egg yolk, specific saturated LPC (18:0), or specific unsaturated LPC (18:1) into standard NGM Petri plates. We placed wild-type embryos on these plates and observed that in all cases embryos developed to adulthood normally. However, animals fed either mixed LPCs from egg yolk or saturated LPC (18:0), but not unsaturated LPC (18:1), generated exclusively dead/arrested embryos (Fig. 4d). These data indicate that an excess of specific saturated LPCs inhibits embryonic development and are consistent with the hypothesis that increased LPC abundance in response to osmotic stress inhibits embryonic development.

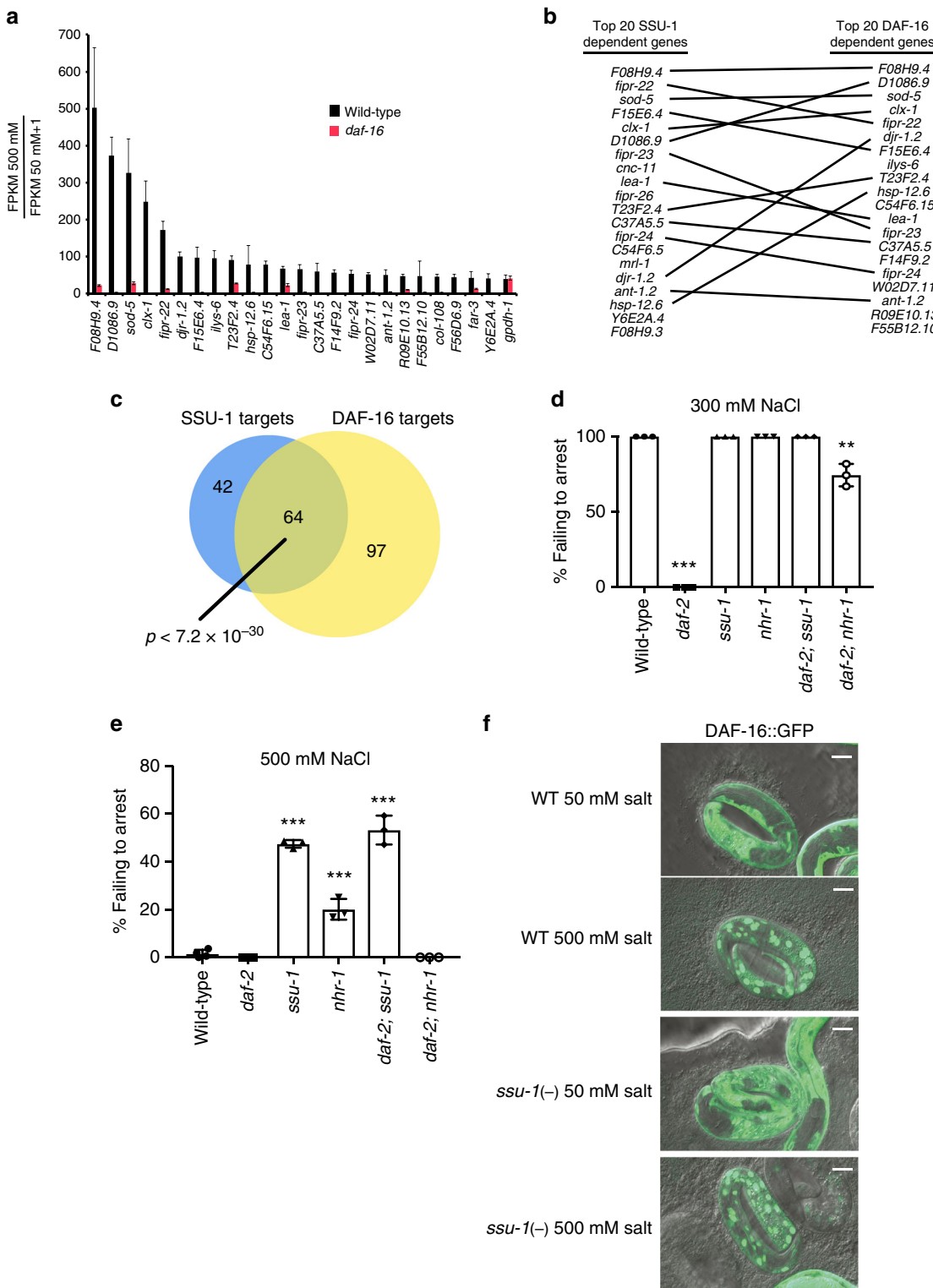

## Discussion

By studying how changes in insulin-like signaling drive a unique transcriptional response and developmental arrest in response to osmotic stress in *C. elegans*, we discovered that an SSU-1 sufotransferase—NHR-1 nuclear hormone receptor signaling pathway acts in concert with an insulin-like signaling pathway that includes the insulin-like peptide INS-3[1], the insulin-like receptor DAF-2[1], and the FOXO transcription factor DAF-16[5] to drive gene expression in response to osmotic stress (Fig. 4d, e). Obese and non-obese

humans show different transcriptional responses in adipose tissue to changes in insulin signaling, and such differences might contribute to or cause pathologies associated with obesity[1,2]. We suggest that differences in neurohormonal signaling mediated by a sulfotransferase—nuclear hormone receptor signaling pathway might underlie the observed differences in gene expression that occur in response to insulin signaling in obese vs. non-obese individuals.

More generally, numerous studies have established that the central nervous system plays an important role in regulating

**Fig. 3** SSU-1 and insulin-like signaling via the FOXO transcription factor DAF-16 function in parallel to regulate gene expression and developmental arrest in response to osmotic stress. **a** Average fold change mRNA expression in the wild-type and *daf-16(m26)* mutant embryos at 500 mM NaCl as compared to 50 mM NaCl measured by RNA-seq (FPKM 500 mM NaCl/FPKM 50 mM NaCl + 1). Shown are the 25 genes that exhibited the greatest increase in expression in wild-type animals at 500 mM NaCl vs. 50 mM NaCl. $n = 3$ replicates. Error bars, s.d. **b** Comparison of the 20 genes exhibiting the largest SSU-1 and DAF-16-dependent increases in expression in response to 500 mM NaCl. **c** Venn diagram of genes with a greater than twofold increase in RNA expression in response to osmotic stress and dependent on SSU-1 (blue) and/or DAF-16 (yellow). *p*-value represents normal approximation to the hypergeometric probability (See Statistics and Reproducibility). **d** Percent of wild-type, *daf-2(e1370)*, *ssu-1(fc73)*, and *nhr-1(n6242)* mutant embryos failing to arrest development in response to 300 mM NaCl. $n > 100$ animals. Error bars, s.d. **e** Percent of wild-type, *daf-2(e1370)*, *ssu-1(fc73)*, and *nhr-1(n6242)* mutant embryos failing to arrest development in response to 500 mM NaCl. $n > 100$ animals. Error bars, s.d. **f** Representative confocal images of DAF-16:: GFP localization after 5 h of exposure to osmotic stress (500 mM NaCl) in the wild-type and *ssu-1(fc73)* mutant embryos. Scale bars, 10 μm. Source data are provided as a Source Data file

insulin sensitivity and specificity[15]. For example, specific neuronal cell populations, such as the agouti-related peptide (AgRP) neurons, can control responses to insulin signaling, including alterations in gene expression in distant tissues[15,16]. However, neither neuronal control of nor cellular responses to insulin signaling have previously been linked to cytosolic sulfotransferase-controlled hormonal signaling. Given the strong conservation of the insulin signaling pathway as a regulator of growth and metabolism throughout metazoan (e.g., much of the insulin signaling pathway was discovered from studies of *C. elegans*), we propose that neuronal cytosolic sulfotransferase activity functions broadly in neurohormonal signaling in mammals and that abnormalities in such signaling underlie a variety of human pathologies known to be caused by abnormalities in insulin signaling, such as obesity and type 2 diabetes.

## Methods

**Strains.** *C. elegans* strains were cultured and maintained at 20 °C unless noted otherwise[17]. The Bristol strain N2 was the wild-type strain. Mutations used were:
 LGI: *daf-16(m26, mu86)*
 LGIII: *daf-2(e1370)*
 LGIV: *ssu-1(fc73, n5883, tm1117, gk266317, gk747222, gk876992, gk319712)*, zIs356 [*Pdaf-16::daf-16a/b-gfp; rol-6*], ayIs7 [*hlh-8::GFP fusion + dpy-20(+)*]
 LGX: *nhr-1(n6217, n6219, n6228, n6231, n6242)*
 Unknown linkage: wuIs57 [*sod-5p::GFP, rol-6(su1006)*]
 Extrachromosomal arrays: nEx2685 [*Ptrx-1::ssu-1::mCherry::unc-54 3'UTR; Punc-122::GFP*], nEx2722 [*Prab-3::nhr-1::mCherry::tbb-2 3'UTR; Punc-122::GFP*], nEx2720 [*Punc-54::nhr-1::mCherry::tbb-2 3'UTR; Punc-122::GFP*], nEx2719[*Pges-1::nhr-1::mCherry::tbb-2 3'UTR; Punc-122::GFP*]

**M-cell division in response to starvation.** Mixed-stage cultures on 10 cm NGM plates were washed from the plates using S-basal and centrifuged. A hypochlorite solution (7:2:1 ddH$_2$O, sodium hypochlorite (Sigma), 5 M KOH) was added to dissolve the animals. Worms were centrifuged after 1.5–2 min in the hypochlorite solution and fresh solution was added. Total time in the hypochlorite solution was 7–10 min. Embryos were washed three times in S-basal buffer (including 0.1% ethanol and 5 ng/μL cholesterol) before final suspension in 5 mL S-basal at a density of 1 worm/μL. Embryos were cultured in a 16 mm glass tube on a tissue culture roller drum at approximately 25 rpm and 21–22 °C. For the M-cell division assay during starvation, the larvae were starved for 7 days before an average of 150 larvae per replicate were examined on a 5% Noble agar slide on a compound fluorescent microscope.

**DAF-16::GFP localization.** Embryos were placed onto Petri plates containing 500 mM NaCl in NGM agar seeded with *E. coli* OP50 for 5 h. Confocal microscopy was performed using a Zeiss LSM 800 instrument. The resulting images were prepared using ImageJ software (National Institutes of Health). Image acquisition settings were calibrated to minimize the number of saturated pixels and were kept constant throughout the experiment.

**Assay for developmental arrest.** Approximately 200 developing eggs from mothers grown at 50 mM NaCl (unless otherwise noted) were collected and placed on standard NGM Petri plates containing varying concentrations of NaCl for 48 h. After 48 h, animals that remained immobile and were not feeding were scored as arrested. Mobile animals that were feeding were scored as developing. Percent failing to arrest was defined as the percent of animals mobile and feeding (unlike animals normally arrested in response to osmotic stress) and includes L1-stage larvae.

**Mutagenesis screening.** Two screens were performed to identify mutants that failed to arrest development in response to osmotic stress. In the screen that identified *ssu-1* (*n5888*, W284Opal) approximately 500 L4 stage wild-type animals were incubated with 20 μL of ethyl methanesulfonate (EMS) (Sigma) in 4 mL of M9 for 4 h at 20 °C, and approximately 20,000 F3 generation embryos were placed onto Petri plates containing 500 mM NaCl in NGM agar and screened for mutants that hatched and were mobile. In the screen that identified the alleles of *nhr-1*, approximately 20,000 L4 stage animals expressing SOD-5::GFP (*wuIs57*) were incubated with 20 μL of EMS in 4 mL of M9 for 4 h at 20 °C, and approximately 200,000 F3 generation embryos were placed onto Petri plates containing 500 mM NaCl in NGM agar and screened for mutants that hatched and were mobile.

**Cloning of *Ptrx-1::ssu-1::mCherry*.** A synthetic *ssu-1* DNA fragment with synthetic introns replacing endogenous introns (endogenous introns are repetitive and could not be synthesized) was obtained from Integrated DNA Technologies using their custom gene-synthesis service. The *ssu-1* fragment was amplified with a 3' 5xGly linker using appropriate primers. The pJDM169 vector containing 1.1 kb of the *trx-1* promoter sequence upstream of the *trx-1* start codon, mCherry and an *unc-54* 3'UTR was obtained from J. Meisel[18]. The *ssu-1* DNA fragment with 5xGly linker was cloned into the pJDM169 using Infusion HD (Clontech) cloning to generate the plasmid pVD100 that contains *Ptrx-1::ssu-1(cDNA) – 5xGly – mCherry::unc-54 3'UTR*.

**Metabolite preparation and quantification.** Approximately 100 μL of embryos were collected by bleaching and placed on standard NGM agar plates. After 3 h embryos were collected in M9, pelleted, and frozen. 100 μL of frozen embryos were resuspended in 600 μL methanol and lysed using a BeadBug microtube homogenizer (Sigma) and 0.5 mm Zirconium beads. After lysis, 300 μL of water and 400 μL of cholorform were added to each sample, and samples were vortexed for 1 min at 4 °C and centrifuged for 10 min at 15,000× *g* at 4 °C. After centrifugation the polar and lipid layers were separated and dried using a SpeedVac concentrator. Liquid chromatography and mass spectrometry were performed as described[19].

**RNA-seq.** Wild-type, *ssu-1(fc73)*, and *daf-16(m26)* embryos were placed on standard NGM plates. After 3 h embryos were collected in M9, lysed using a BeadBug microtube homogenizer (Sigma) and 0.5 mm Zirconium beads (Sigma), and RNA was extracted using the RNeasy Mini kit (QIAGEN). RNA integrity and concentration were assessed using a Fragment Analyzer (Advanced Analytical), and libraries were prepared using a Illumina NeoPrep RNAseq kit. Libraries were loaded for paired-end sequencing using the Illumina NextSeq500. Seventy-five-nucleotide paired-end sequencing reads were mapped against the *C. elegans* genome assembly ce10 using RSEM v. 1.2.15, with Bowtie v. 1.0.1 for read alignment (flags --paired-end -p 6 --bowtie-chunkmbs 1024 --forward-prob 0, for strand-specific libraries)[20,21]. Expected read counts per gene were retrieved and, after rounding counts to the nearest integer, used to perform differential gene expression analysis with DESeq2 in the R v. 3.2.3 statistical environment. Sequencing library size was estimated for each library to account for differences in sequencing depth and complexity among libraries, as well as gene-specific count dispersion parameters. Gene identifiers were obtained from Wormbase version WS235.

**Psod-5::GFP images.** For Fig. 2b, c embryos were placed onto Petri plates containing either 50 mM or 500 mM NaCl in NGM agar. Arrested L1-stage animals at 500 mM NaCl were imaged after 24 h. L1-stage animals at 50 mM NaCl were imaged after 5 h to control for staging. For Fig. 2f dauer larvae were collected 7 days after starvation on plates containing 500 mM NaCl. For Fig. 2g embryos were placed onto Petri plates containing 50 mM NaCl in NGM agar and imaged after 24 h. Arrested L1-stage animals at 500 mM NaCl were imaged after 24 h For Figs. 2b, c, 3f confocal microscopy was performed using a Leica SP8 instrument. For Fig. 2f, g confocal microscopy was performed using a Zeiss LSM 800 instrument. The resulting images were prepared using ImageJ software (National Institutes of Health). Image acquisition settings were calibrated to minimize the number of saturated pixels and were kept constant throughout each experiment.

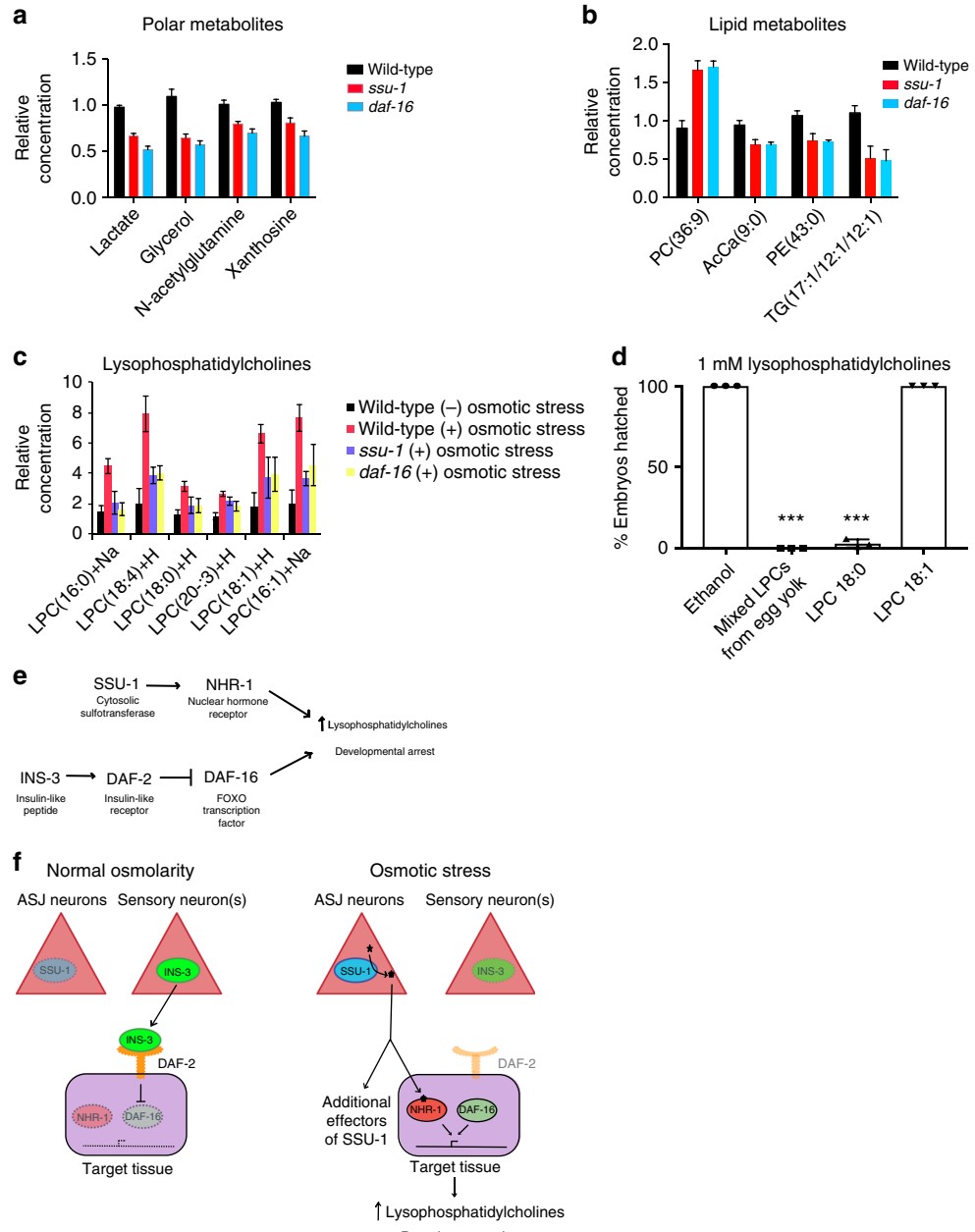

**Fig. 4** SSU-1 and DAF-16 regulate lysophosphatidylcholine levels in response to osmotic stress. **a** Relative levels of polar metabolites exhibiting a statistically significant ($p < 0.01$) change in levels in *ssu-1(fc73)* and *daf-16(m26)* mutant embryos. Metabolites were normalized to the levels of histidine. $n = 3$ replicates. Error bars, s.d. **b** Relative levels of lipid metabolites exhibiting a statistically significant ($p < 0.01$ two-tailed *t*-test) change in levels in both *ssu-1(fc73)* and *daf-16(m26)* mutant embryos. Metabolites were normalized to the levels of total lipid. $n = 3$ replicates. Error bars, s.d. PC phosphatidylcholine, AcCa acylcarnitine, PE phosphatidylethanolamine, TG triglyceride. **c** Relative levels of lysophosphatidylcholine (LPC) metabolites in the wild-type and *ssu-1(fc73)* and *daf-16(m26)* mutant embryos. Metabolites shown exhibited a statistically significant ($p < 0.01$ two-tailed *t*-test) and greater than twofold increase in abundance in response to osmotic stress (300 mM NaCl) and required both SSU-1 and DAF-16. Metabolites were normalized to the levels of total lipid. $n = 3$ replicates. Error bars, s.d. **d** Percent of wild-type embryos from parents fed 1 mM lysophosphatidylcholines (LPCs) that hatch. Error bars, s.d. **e** Diagram of the genetic pathway regulating *C. elegans* development in response to osmotic stress. This diagram is based on the results described here and the insulin-like signaling pathway regulating developmental arrest in response to osmotic stress described previously by Burton et al.[5]. **f** Model for how *C. elegans* regulates development in response to osmotic stress. This model is based on the results described here and the results described previously by Burton et al.[5] Source data are provided as a Source Data file

**Cloning of *nhr-1* rescuing transgenes**. *R09G11.2c* was amplified from *C. elegans* wild-type cDNA and used in all vectors generated for tissue-specific expression of *nhr-1* cDNA. Infusion HD (Clontech) cloning was used to clone individual promoter fragments with *nhr-1* cDNA, 5xGly linker, mCherry and *tbb-2 3'UTR* to generate plasmids containing the generic sequence *Promoter::nhr-1 cDNA – 5xGly – mCherry::tbb-2 3'UTR*. The promoter fragments of the genes *ges-1* (in pVD104), *unc-54* (in pVD105), and *rab-3* (in pVD106) used for tissue-specific expression of

*nhr-1* cDNA contain 2.9, 1.9, 1.4 and 1.3 kb, respectively, of sequence upstream of each of these genes' start codons.

**Germline transformations**. Extrachromosomal arrays were generated by injecting the corresponding plasmid and co-injection marker into the gonad of one-day old adults at the specified concentrations. Animals in the F1 progeny carrying the

resulting extrachromosomal arrays were individually picked onto separate Petri plates and examined for transmission to progeny. Plates with progeny carrying transmitted arrays were maintained as independent transgenic lines. *nEx2685* was generated by injecting pVD100 at 20 μg/mL and the co-injection marker plasmid pQZ22 at 50 μg/mL into ssu-1(fc73); *wuIs57* animals. *nEx2719*, *nEx2720*, and *nEx2722* were generated by injecting pVD104, pVD105, and pVD106, respectively, at 20 μg/mL along with the co-injection marker plasmid pQZ22 at 50 μg/mL for each transgene into *wuIs57; nhr-1(n6242)* animals.

**Lysophosphatidylcholine (LPC) feeding and hatching**. Mixed LPCs from egg yolk (Sigma L4129), LPC (18:0) (Sigma L2131), and LPC (18:1) (Sigma L1881) were resuspended in ethanol and added to standard NGM Petri plates at a final concentration of 1 mM. These plates were then seeded with *E. coli* OP50. Wild-type embryos were placed on plates containing either ethanol alone or LPCs in ethanol. Animals were allowed to grow for 72 h at room temperature. After 72 h, embryos were collected from adult animals and placed on new petri plates containing LPCs for 24 h, and the fraction of embryos hatched was counted.

**Statistics and reproducibility**. Two-tail *t*-tests were used to compare all samples that reflect percentages of populations or populations of animals. For Fig. 3c the *p*-value was calculated using a normal approximation to the hypergeometric probability, as in http://nemates.org/MA/progs/representation.stats.html. No statistical method was used to predetermine sample size. The experiments were not randomized. The investigators were not blinded to allocation during experiments and outcome assessment. Statistics source data available as a Source Data file.

## Data availability
All relevant data are available from the authors on reasonable request and/or are included with the manuscript. RNA-seq data are available using the GEO accession number GSE111074. The source data underlying Figs. 1b, d, 2e, g, 3d, and e are provided as a Source Data file.

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

## Acknowledgements
We thank David Gems and the Caenorhabditis Genetic Center, which is funded by the NIH National Center for Research Resources (NCRR), for strains; N. An for strain management; and A. Doi, and A. Corrionero for helpful discussions. H.R.H., K.B.B., and N.O.B. were supported by NIH grant GM024663, and N.O.B. was also supported by NSF grant 1122374 and a Next Generation Fellowship from the Center for Trophoblast Research. V.K.D. was a Howard Hughes Medical Institute International Student Research fellow. L.R.B. and R.E.W.K. were supported by NIH grant GM117408. H.R.H. is an Investigator of the Howard Hughes Medical Institute.

## Author contributions
N.O.B., R.E.W.K., L.R.B., and H.R.H. designed the experiments and analyzed the data. N.O.B., V.K.D, K.B.B, and R.E.W.K performed the experiments. N.O.B. and H.R.H wrote the manuscript.

## Additional information

**Competing interests:** The authors declare no competing interests.

