## [Transparent Peer Review File · Nature Communications]

Reviewers' comments:

Reviewer #1 (Remarks to the Author):

Insulin and insulin-like signaling (IIS) have profound effects on a broad range of physiological and pathological processes in various species. In this manuscript, the authors performed a nice genetic study to explore the mechanisms by which IIS signaling elicits distinct responses to different environmental stresses. They found that the cytosolic sulfotransferase SSU-1 in ASJ neurons is specifically required for osmotic stress-induced larva arrest. They showed that the nuclear hormone receptor NHR-1 is required for SSU-1 in ASJ neurons to mediate transcriptional responses to osmotic stress but not starvation. They conclude that signals from SSU-1 in ASJ neuron most likely functions in parallel to reduced IIS signaling to modulate development arrest induced by osmotic stress in *C. elegans*.

Overall, this is a nice study. Well designed and executed. As the objective of the study is to identify genes important for IIS to elicit distinct responses to osmotic vs. starvation stress, it is a bit disappointing that the authors only performed screens to identify the genes important for osmotic stress resistance but not for starvation resistance. Otherwise, it would be a master piece of work. There are several questions that need to be addressed.

- 1, The authors performed the M-cell division assay and concluded that SSU-1 might not be required for starvation-induced developmental arrest. How about the percentage of larva-arrested *ssu-1* mutant worms under starvation?
- 2, The failure to rescue the phenotype of *nhr-1* mutant worms even with its endogenous promoter is quite surprising. More explanations and discussions are required.
- 3, Is NHR-1 required for reduced IIS-mediated responses to osmotic stress?
- 4, There are several insulin-like peptides expressed in ASJ neurons. Does SSU-1 have any effect on the expression and release of those insulins?
- 5, How does the neuronal activity of ASJ neurons change in response to osmotic stress?
- 6, According to the model proposed by the authors, NHR-1 and DAF-16 are acting downstream of SSU-1 and DAF-2, respectively. It makes more sense to compare the transcriptional responses mediated by NHR-1 and DAF-16 rather than SSU-1 and DAF-16.

Reviewer #2 (Remarks to the Author):

This elegant manuscript from Burton et al, advances our knowledge of the neural hormone responses to stress. The paper is important as provides mechanistic details into response pathways to osmotic stress and identifies a new player in the response, a cytosolic sulfotransferase (SSU-1) that they demonstrate acts in a pair of neurons to co-ordinate downstream responses. Most of the findings derive from well-designed genetic experiment and the authors interpretation of the results are appropriate. The authors continue to explore mechanisms at play by a combination of transcriptomic and metabolomic methods and identify lysophosphatidylcholine metabolism as being altered.

I have the following concerns that should be addressed prior to publication –

- 1) Why is the RNA seq data and some other gene expression data collected from embryos? The phenotyping is conducted at later stages. I'm sure there is a good reason for this, but if not it seems like an important disconnect.
- 2) I can't find the statistical analysis of the various transcription profile comparisons. Many statements are made about the number of overlapping genes, but I don't see statistical significance assigned to each. This is of course necessary.

3) The alteration in lysophosphatidylcholine metabolism is interesting but there is no attempt to address the mechanistic significance. The manuscript ends abruptly on this discovery where one might expect experiments addressing causality.

Reviewer #3 (Remarks to the Author):

This interesting manuscript aims to address the mechanisms that enable insulin-like signalling to generate distinct states of developmental arrest due to different stresses. The principal results reveal the role of the *ssu-1* sulfotransferase and its connection to the *nhr-1* nuclear hormone transcription factor in regulating developmental arrest in response to osmotic stress. This study also uncovers interactions between the *ssu-1/nhr-1* pathway and the insulin-like signalling pathway, based on their roles in this developmental arrest and their effects on transcriptional and metabolomic responses to osmotic stress. The results support the conclusion that *ssu-1/nhr-1* works together with the insulin-like pathway to specify developmental arrest and other responses to osmotic stress.

Major Issues

1. The physiological relevance of the transcriptomic, and metabolomic readouts is not apparent. For example, whether *sod-5* or lysophosphatidylcholines mediate the arrest process, or is part of a wider response that aids survival under high-salt is unclear.
2. The authors show that *sod-5* expression is responsive to both osmotic stress (Fig. 2b, L1 arrest in high salt) and starvation (Fig. 2f, dauer arrest due to starvation), but due to technical reasons, rely on the expression in starvation-generated dauers to test the cell-autonomy of *nhr-1*. This weakens the idea that *nhr-1* is specific to osmotic stress but not starvation.
3. The narrative about *ssu-1* providing specificity to insulin-like signalling is not strongly supported by the results presented, as the results largely focus on the osmotic stress response. I suggest that the authors either shift the narrative to the regulation of osmotic stress-mediated arrest, or provide additional data about starvation responses (e.g. compare *ssu-1*-dependent transcriptional responses in starvation versus osmotic stress) to reinforce the point about specificity in insulin-like signalling.

Minor Issues

Page 7 line 180: for clarity, briefly indicate that rescue was not observed, possibly due to overexpression of *nhr-1* and then direct readers to the supplemental discussion.

If the authors think that overexpression is the main obstacle to rescuing *nhr-1*, they can use single copy integrants of the rescuing transgene to address this problem.

Introducing *ins-3* will make Fig. 4e more clear.

The figures look small and should be enlarged. In particular, the gene names in Fig. 2a are not legible without zooming in.

Reviewers' comments:

Reviewer #1 (Remarks to the Author):

Insulin and insulin-like signaling (IIS) have profound effects on a broad range of physiological and pathological processes in various species. In this manuscript, the authors performed a nice genetic study to explore the mechanisms by which IIS signaling elicits distinct responses to different environmental stresses. They found that the cytosolic sulfotransferase SSU-1 in ASJ neurons is specifically required for osmotic stress-induced larva arrest. They showed that the nuclear hormone receptor NHR-1 is required for SSU-1 in ASJ neurons to mediate transcriptional responses to osmotic stress but not starvation. They conclude that signals from SSU-1 in ASJ neuron most likely functions in parallel to reduced IIS signaling to modulate development arrest induced by osmotic stress in *C. elegans*.

Overall, this is a nice study. Well designed and executed. As the objective of the study is to identify genes important for IIS to elicit distinct responses to osmotic vs. starvation stress, it is a bit disappointing that the authors only performed screens to identify the genes important for osmotic stress resistance but not for starvation resistance. Otherwise, it would be a master piece of work. There are several questions that need to be addressed.

1, The authors performed the M-cell division assay and concluded that SSU-1 might not be required for starvation-induced developmental arrest. How about the percentage of larva-arrested *ssu-1* mutant worms under starvation?

daf-16 mutants are considered not to arrest development in response to starvation because certain cells that in wild-type animals stop dividing, such as the M cell and the seam cells, continue to divide. Superficially, starved *daf-16* larvae are indistinguishable from arrested larvae. It is for this reason that we used M cell division as an assay for animals that failed to arrest development in response to starvation.

As we discuss in response #3 to Reviewer 3, M cell division is only one aspect of developmental arrest in response to starvation and that while our data are consistent with a model in which SSU-1 is not required for developmental arrest in response to starvation (Fig. 1c), it is possible that SSU-1 plays a partial role in some aspects of arrest in response to starvation. For this reason, we have modified the text in several places to focus our presentation more on the regulation of the arrest in response to osmotic stress than on a comparison between the arrest in response to osmotic stress and to starvation, as suggested by Reviewer 3 #3 (abstract line 54-55 and main text lines 133-135, 206-209, and 318-324).

2, The failure to rescue the phenotype of *nhr-1* mutant worms even with its endogenous promoter is quite surprising. More explanations and discussions are required.

We have added text to the Supplemental Discussion to discuss this result (see lines 704-722). Briefly, there are multiple possible reasons this extrachromosomal transgene might fail to rescue. One possibility is that transgene-driven overexpression of NHR-1 in some tissues or in embryos results in arrest/lethality. If so, the only animals we recovered from our injections of *nhr-1* under the control of its endogenous promoter might have silenced the transgene or incorrectly expressed *nhr-1*. Alternatively, it is possible that the extrachromosomal array does not express sufficiently early in embryonic development or in the correct tissues to rescue the effects of loss of NHR-1 in response to osmotic stress because the promoter fragment we

used does not capture the entire *nhr-1* promoter. We used a 3 kb promoter for *nhr-1*, but the next closest gene to *nhr-1* is > 25 kb away and a longer fragment of the promoter might be needed to drive proper gene expression. A third possibility is that the isoform we expressed (R09G11.2c) is not the isoform that is functional in embryos to control this particular response to osmotic stress.

3, Is NHR-1 required for reduced IIS-mediated responses to osmotic stress?

We agree that testing whether mutations in *nhr-1* can suppress mutations in *daf-2* (insulin-like receptor) with respect to developmental arrest in response to osmotic stress is an important experiment. We performed this experiment and found that, as expected, like *ssu-1*, *nhr-1* is required in *daf-2* mutants for developmental arrest in response to osmotic stress at 300 mM NaCl (new Fig. 3d).

However, unlike *ssu-1*, *nhr-1* is not required in *daf-2* mutants to arrest development at 500 mM NaCl (new Fig. 3e). This new result suggests that NHR-1 is not the only downstream effector of SSU-1. Specifically, the finding that SSU-1, but not NHR-1, is required for *daf-2* mutants to arrest development at 500 mM NaCl indicates that SSU-1 has an NHR-1-independent function. Thus SSU-1 likely functions similarly to cytosolic sulfotransferases in other organisms, e.g., in humans, individual sulfotransferases commonly sulfonate multiple targets -- SULT2A1 sulfonates testosterone, dehydroepiandrosterone, androsterone, and pregnenolone -- to modify the downstream activity of multiple hormone receptors.

4, There are several insulin-like peptides expressed in ASJ neurons. Does SSU-1 have any effect on the expression and release of those insulins?

ins-1, *ins-3*, *ins-4*, *ins-6*, *ins-9*, and *ins-32* have been reported to be expressed in ASJ. However, we observed no changes in the expression of these insulin-like peptides using RNA-seq to study embryos +/- osmotic stress (Supplemental Table 1). In addition, our genetic data indicate that mutations in *ssu-1* can suppress mutations in the insulin-like receptor DAF-2 (Fig. 3d and 3e) and that mutations in *ssu-1* do not affect the translocation of DAF-16 into the nucleus in response to osmotic stress (Fig. 3f). Both of these results would not be expected if SSU-1 functioned by regulating the expression or release of insulin-like peptides. Thus, we conclude that SSU-1 functions in parallel to insulin-like signalling to control *C. elegans* response to osmotic stress and likely does not modify the expression or release of insulin-like peptides.

5, How does the neuronal activity of ASJ neurons change in response to osmotic stress?

Our data support a model in which SSU-1 activity in the ASJ sensory neurons changes in response to osmotic stress. It is possible that other activities of ASJ, such as calcium signalling, also change in response to osmotic stress. We have not assayed other activities of ASJ and do not believe that such studies are necessary for the conclusions of this manuscript.

6, According to the model proposed by the authors, NHR-1 and DAF-16 are acting downstream of SSU-1 and DAF-2, respectively. It makes more sense to compare the transcriptional responses mediated by NHR-1 and DAF-16 rather than SSU-1 and DAF-16.

We agree with that this comparison would be the more natural comparison based on our original model. However, we have updated our model in light of our finding that NHR-1 is

not the only downstream effector of SSU-1 (new Fig. 3D and 3E - see response to Reviewer 1- #3). Based on these new data, which indicate that SSU-1 might modify the activity of multiple downstream nuclear hormone receptors (including NHR-1), we believe that comparing the transcriptional responses mediated by SSU-1 and DAF-16 is a broader and more informative comparison for the understanding of which genes are regulated by both insulin-like signalling (via DAF-16) and SSU-1-mediated signalling (via NHR-1 and as yet unknown effectors).

Reviewer #2 (Remarks to the Author):

This elegant manuscript from Burton et al, advances our knowledge of the neural hormone responses to stress. The paper is important as provides mechanistic details into response pathways to osmotic stress and identifies a new player in the response, a cytosolic sulfotransferase (SSU-1) that they demonstrate acts in a pair of neurons to co-ordinate downstream responses. Most of the findings derive from well-designed genetic experiment and the authors interpretation of the results are appropriate. The authors continue to explore mechanisms at play by a combination of transcriptomic and metabolomic methods and identify lysophosphatidylcholine metabolism as being altered.

I have the following concerns that should be addressed prior to publication –

1) Why is the RNA seq data and some other gene expression data collected from embryos? The phenotyping is conducted at later stages. I'm sure there is a good reason for this, but if not it seems like an important disconnect.

There are two reasons we performed RNA-seq studies using embryos. The first is to control for staging. Mutants that fail to arrest development in response to osmotic stress hatch and begin feeding/developing, whereas wild-type animals arrest development and do not begin feeding/developing. We believe that this difference after hatching would introduce differences in gene expression that are a less direct consequence of osmotic stress, and we wanted to eliminate any such differences.

The second reason is that we believe that the decision to arrest development is likely made during late embryonic development, before hatching. In support of this hypothesis, (1) we observed that some animals arrest development before hatching, and (2) we previously showed that the transcriptional response to osmotic stress is activated in embryos (Burton et al., 2017).

For these two reasons, we chose to perform RNA-seq studies using embryos and indeed observed the activation of gene expression in response to osmotic stress in embryos and showed that this activation of gene expression is dependent on SSU-1 and DAF-16.

2) I can't find the statistical analysis of the various transcription profile comparisons. Many statements are made about the number of overlapping genes, but I don't see statistical significance assigned to each. This is of course necessary.

We agree and thank Reviewer 2 for this suggestion. The statistical analysis has been added.

3) The alteration in lysophosphatidylcholine metabolism is interesting but there is no attempt to address the mechanistic significance. The manuscript ends abruptly on this discovery

where one might expect experiments addressing causality.

We have since tested if supplementing the *C. elegans* diet with various lysophosphatidylcholines (LPCs) affects embryonic or larval development. Interestingly, we observed that the addition of specific saturated, but not unsaturated, LPCs that change in abundance in response to osmotic stress to the *C. elegans* diet caused embryos to arrest development. By contrast, these same LPCs did not appear to affect larval development. Given these results, we hypothesize that specific saturated LPCs regulate embryonic development in response to osmotic stress. These data have been added to Figure 4.

Reviewer #3 (Remarks to the Author):

This interesting manuscript aims to address the mechanisms that enable insulin-like signalling to generate distinct states of developmental arrest due to different stresses. The principal results reveal the role of the *ssu-1* sulfotransferase and its connection to the *nhr-1* nuclear hormone transcription factor in regulating developmental arrest in response to osmotic stress. This study also uncovers interactions between the *ssu-1/nhr-1* pathway and the insulin-like signalling pathway, based on their roles in this developmental arrest and their effects on transcriptional and metabolomic responses to osmotic stress. The results support the conclusion that *ssu-1/nhr-1* works together with the insulin-like pathway to specify developmental arrest and other responses to osmotic stress.

Major Issues

1. The physiological relevance of the transcriptomic, and metabolomic readouts is not apparent. For example, whether *sod-5* or lysophosphatidylcholines mediate the arrest process, or is part of a wider response that aids survival under high-salt is unclear.

As described in Reviewer 2 Response #3, we have now added data supporting the hypothesis that LPCs have a physiological role in the response to osmotic strength.

Concerning the transcriptomic data, we note that previous studies demonstrated that several of the genes that exhibit increased expression in response to osmotic stress are required for survival and development in response to osmotic stress, e.g., *gpdh-1* and *gpdh-2* (Lamatina et al., 2006; Burton et al., 2017). Thus it seems likely that additional genes that exhibit increased expression in response to osmotic stress also enhance survival in response to osmotic stress. We have revised the text to include this information (see lines 165-170).

We have not observed any effect of the loss of *sod-5* on the response of *C. elegans* to osmotic stress and for this reason use *sod-5* expression only as a marker of the osmotic-stress response. It is possible that SOD-5 functions redundantly with other genes that exhibit increased expression in response to osmotic stress. For example, the expression of another superoxide dismutase gene, *sod-3*, is also significantly increased in response to osmotic stress, and SOD-3 and SOD-5 might function redundantly in the response to osmotic stress.

2. The authors show that *sod-5* expression is responsive to both osmotic stress (Fig. 2b, L1 arrest in high salt) and starvation (Fig. 2f, dauer arrest due to starvation), but due to technical

reasons, rely on the expression in starvation-generated dauers to test the cell-autonomy of *nhr-1*. This weakens the idea that *nhr-1* is specific to osmotic stress but not starvation.

We agree that our data indicate that SSU-1/NHR-1 signalling plays a role in dauer-arrested animals, including in dauer animals arrested in response to starvation. We have revised the text to be more specific in describing this observation and to raise the possibility that SSU-1/NHR-1 signalling might play a broader role in *C. elegans* stress responses than in only osmotic stress (lines 206-209). Nonetheless, unlike animals that arrest as L1s in response to starvation, dauer-arrested animals are resistant to a wide-array of stresses, including osmotic stress. Thus it is possible that SSU-1/NHR-1 signalling in dauer-arrested animals is simply an aspect of dauer resistance to osmotic stress.

3. The narrative about *ssu-1* providing specificity to insulin-like signalling is not strongly supported by the results presented, as the results largely focus on the osmotic stress response. I suggest that the authors either shift the narrative to the regulation of osmotic stress-mediated arrest, or provide additional data about starvation responses (e.g. compare *ssu-1*-dependent transcriptional responses in starvation versus osmotic stress) to reinforce the point about specificity in insulin-like signalling.

We agree and have revised the text to focus on the osmotic stress response (abstract line 54-55 and main text lines 133-135, 206-209, and 318-324).

Minor Issues

Page 7 line 180: for clarity, briefly indicate that rescue was not observed, possibly due to overexpression of *nhr-1* and then direct readers to the supplemental discussion.

The text has been revised to address this concern (lines 182-183).

If the authors think that overexpression is the main obstacle to rescuing *nhr-1*, they can use single copy integrants of the rescuing transgene to address this problem.

As described in Reviewer 1 Response #2, overexpression of NHR-1 is only one of a number of possible obstacles, including the possibility that the promoter fragment we used is not being sufficient to drive NHR-1 expression in the right place/at the right time. We have revised the Supplemental Discussion to consider these possibilities.

Introducing *ins-3* will make Fig. 4e more clear.

We agree and thank Reviewer 3 for this suggestion. We have revised the text to introduce *INS-3* (lines 318-323).

The figures look small and should be enlarged. In particular, the gene names in Fig. 2a are not legible without zooming in.

The figures have been updated and enlarged.

REVIEWERS' COMMENTS:

Reviewer #1 (Remarks to the Author):

The authors have addressed many of my comments. It is a bit disappointing that the authors only performed screens to identify the genes important for osmotic stress resistance but not for starvation resistance, which kind of limits the impact of the work. Nevertheless, their work on osmotic stress resistance does advance the field, and I therefore have no problem to see it published in NCOMMS.

Reviewer #2 (Remarks to the Author):

The authors have responded adequately to my questions. It is a more complete paper and worthy of publication.

Reviewer #3 (Remarks to the Author):

I thank the authors for their efforts. They have addressed the issues that I raised, and have strengthened the manuscript tremendously with their new data.

I took note of point #5 made by reviewer 1 regarding the activity of ASJ in response to osmotic stress. This experiment was reported in Zaslaver et al PNAS 2014 in Figure 2; they show that ASJ is responsive to increases in NaCl, but not to osmotic stress. This result turns out to be important in that it suggests that signals initiated in ASJ are in response to high NaCl, rather than to general osmotic stress.

While this result does not in any way change the interest or novelty of this manuscript, I believe it would be more conservative for the authors to describe the ssh/1nhr-1 pathway as responding to high salt rather than general osmotic stress.